# Faster Deep Inertial Pose Estimation with Six Inertial Sensors

**DOI:** 10.3390/s22197144

**Published:** 2022-09-21

**Authors:** Di Xia, Yeqing Zhu, Heng Zhang

**Affiliations:** School of Computer and Information Science, Southwest University, Chongqing 400700, China

**Keywords:** pose estimation, SRU, sparse inertial sensor, occlusion

## Abstract

We propose a novel pose estimation method that can predict the full-body pose from six inertial sensors worn by the user. This method solves problems encountered in vision, such as occlusion or expensive deployment. We address several complex challenges. First, we use the SRU network structure instead of the bidirectional RNN structure used in previous work to reduce the computational effort of the model without losing its accuracy. Second, our model does not require joint position supervision to achieve the best results of the previous work. Finally, since sensor data tend to be noisy, we use SmoothLoss to reduce the impact of inertial sensors on pose estimation. The faster deep inertial poser model proposed in this paper can perform online inference at 90 FPS on the CPU. We reduce the impact of each error by more than 10% and increased the inference speed by 250% compared to the previous state of the art.

## 1. Introduction

Three-dimensional human pose estimation is significant in bio-mechanical, movie production, and gaming. Currently, the most popular approach to solving the human pose estimation task is to use vision-based deep learning algorithms. Vision-based systems typically solve the occlusion problem using multiple cameras, which makes the system complex, large and expensive. Another high-precision strategy is to use optical marker-based human pose estimation by wearing several optical markers on the human body and recording the optical signals through cameras to accomplish 3D human pose estimation, such as the widely used Vicon. However, such systems need expensive infrastructure and intrusive equipment, which makes them unsuitable for consumer-level use, and optical marker-based algorithms do not address the occlusion problem. In contrast to vision-based systems, motion capture utilizing sensors is environment-independent and occlusion-unaware. These qualities make it more appropriate for client-level use. Commercial inertial motion capture systems use 17 IMUs to capture full-body movements, which is very inconvenient and expensive for users. This severely limits the entry of free motion capture systems into the consumer sector. Reconstructing full-body motion from a small number of inertial sensors has now been shown to be feasible. The work of Von et al. [1] shows that it is feasible to reconstruct the human motion pose using only six IMUs. However, as an RNN-based approach, the entire sequence needs to be accessed, and the processing time is extended. Huang et al. [2] implemented an online sparse inertial reconstruction system using a biRNN to estimate the pose distribution. TransPose [3] achieves a more accurate estimation by using biRNN through six IMUs and accomplishing the regression of the global displacement of the human body using just six IMUs. However, TransPose divides the pose prediction problem into many subtasks to increase the accuracy of posture estimation. Each subtask utilizes a distinct training strategy, resulting in an unnecessarily complicated model. Based on DIP and TransPose, we optimized the neural network model structure, simplify the neural network model and training approach, and increased the online performance of the system. We propose the FDIP: faster deep inertial poser, which utilizes the Bi-directional SRU to replace the classical BiRNN and dramatically improves the training speed, with 2.5 times faster inference, 3° lower online prediction SIP error, and 1.5° lower angular error compared with the competitor TransPose’s method.

In this paper, Section 2 presents the related work on pose estimation. Section 3 describes the human parameter model and dataset used. Section 4 presents the FDIP model, which includes the normalization method, the whole model structure, and the fine-tuning method. Section 5 gives the quantitative and qualitative evaluation results of FDIP on the DIP-IMU dataset and does ablation experiments on the proposed method to prove the method’s validity. Section 6 discusses the application limitations and future work of FDIP, and finally, in Section 7, the conclusions of the whole FDIP method are given.

Contributions:

1. The neural network structure was improved using one-stage pose regression without positional supervision to improve the training and inference speed and provide better results.

2. We optimized the method of synthesizing the inertial dataset to be closer to the real inertial dataset.

3. We performed ablation studies on the previous methods presented by SIP [1], DIP, and TransPose, and confirmed that our proposed method achieves SOTA.

## 2. Related Work

### 2.1. Camera-Based Approach

Vision-based human pose estimation is the most prevalent approach. Cameras can capture human motion, and multiple cameras are generally deployed to solve the occlusion problem. Liu et al. [4] utilized rich temporal features between video frames to assist keypoint recognition, encoding the keypoint spatiotemporal context in providing an adequate search space. Yu et al. [5] considered that the perspective effect produced by projection in traditional cameras varies due to the different global positions in the image. To solve the pose reconstruction task, which requires a large amount of 3D real data, Kocabas et al. [6] proposed EpipolarPose, a self-supervised learning method for 3D human pose estimation that does not require any 3D real data or camera extrinsic data. Pavllo et al. [7] devised a simple and effective semi-supervised training strategy to leverage unlabeled video data to quickly predict 3D poses in films using a fully convolutional model based on dilated temporal convolution over 2D keypoints. Rempe et al. [8] proposed an expression model in the form of a conditional variational autoencoder that learns the sequence distribution of pose changes at each step and enables the robust estimation of plausible poses from ambiguous observations. For different commonly used monocular cameras, egocentric 3D body pose estimation is performed on monocular images captured by a downward-looking fisheye camera mounted on the edge of a VR headset. The resulting images have a unique visual appearance, characterized by severe self-occlusion and substantial perspective distortions, resulting in considerable differences in the resolution of the upper and lower body, for which Tome et al. [9] conceived a unique encoder–decoder structure with a dual-branch decoder is proposed, mainly to solve the variable uncertainty of 2D joint positions. Qiu et al. [10] proposed a method for recovering an absolute 3D human pose from multiview images by adding multiview geometric priors to the model. This consists of two separate steps: the estimation of 2D poses in the multiview image and the recovery of 3D poses from multiview 2D poses. All these methods require a large number of computational resources to accurately estimate the attitude. To solve this problem, Hung-Cuong et al. [11] proposed a fast, unified end-to-end model for estimating the 3D human pose, called YOLOv5-HR-TCM ( YOLOv5-HRet-Temporal Convolution Model). Hung-Cuong et al. also applied the YOLOv5-HR-TCM to the assessment and scoring of artistic gymnastics and training and dance assessments.

### 2.2. Based on Visual-Inertial Fusion

Von et al. [12] demonstrated a single handheld camera and a set of connected limb inertial measurement units to estimate accurate 3D outdoor poses. The authors proposed associating 2D pose detection in each image with the corresponding IMU-equipped person and then using a continuous optimization framework to optimize the statistical body model pose, camera pose association, and heading drift. Gilbert et al. [13] used the inertial measurement unit (IMU) sensor fusion data from multi-view video (MVV) to use a multichannel 3D convolutional neural network from visual occupancy and semantic 2D pose estimation to learn to pose embedding from discrete volumetric probabilistic visual shells from MVV. Trumble et al. [14] trained a symmetric convolutional autoencoder with double loss to force learning to encode the potential representations of skeletal joint positions and learn the depth representations of volumetric body shape, resulting in a sparse set of the simultaneous estimation of 3D human pose and body shape in wide baseline camera views based on the method of Qiu et al. [10]. Zhang et al. [15] proposed to estimate 3D human pose from multi-view images and some IMUs connected to human limbs. It first detects a 2D pose from two signals and then lifts it to 3D space.

### 2.3. Based on Inertial Measurement Unit

Commercial inertial motion capture systems, such as Xsens [16], which uses 17 wearable IMUs, can capture all human motion. However, wearing a large number of sensors is invasive for the user. Therefore, it is required to reconstruct human pose from sparse IMU data. New study findings leverage acceleration or orientation and completely utilize IMU to obtain improved accuracy. Von et al. [1] developed a way to solve human motion with only six IMUs. It must run offline as an iterative optimization-based approach, making it impractical to execute in real-time applications. Huang et al. [2] proposed employing just six IMUs to reconstruct full-body positions in real time. The authors deployed a bidirectional recurrent neural network [17] to directly train IMU measurements to body joint rotations. DIP delivers good capture quality and efficiency, but there is still an opportunity for additional development. In the current state of the art, Yi et al. [3] decomposed the human pose estimation task into steps, i.e., estimating joint positions as intermediate representations and regressing joint angles, considerably enhancing the task accuracy. At the same time, the authors suggested a new approach to estimating the overall translation of the wearer without any direct distance measurement, which is a mix of support-foot-based derivation and network-based prediction. We prove that multi-stage tasks will lead to complicated models and slow the inference speed. We used the IMU measurement data to directly forecast the joint’s rotation. The input of DIP [2] removes the acceleration and rotation data of the root node. Although we used normalization for the other sensors relative to the cross-joint, the acceleration information of the cross-joint has a tremendous impact on predicting the final pose. Unlike DIP [2], we retain the data of the root node and anticipate the 6d rotation of the joint. The final test results reveal that our prediction SIP angle error is decreased by 16%, and online inference is 250% quicker on the CPU than TransPose [3].

## 3. Implement

In this section, we introduce our dataset and data pre-processing methods.

### 3.1. Human Parametric Model

We used the same method as DIP [2] to apply the standard parametric human model SMPL [18] to our work. Since we used the AMASS dataset for training, and the AMASS dataset gives the pose parameters applied to the SMPL model, we continued this treatment and used the SMPL model for pose estimation. Moreover, this parameterization-based model is easier to train.

The SMPL model is defined as:(1)$$M\left(\theta \right)=W\left(\bar{T},J,\theta ,w\right).$$
where *T* denotes the preset posture data, i.e., the posture when all joints are rotated to a zero matrix; *J* denotes the 24 body joints; θ denotes the posture parameter represented by the axis angle; *w* denotes the blend weight; and *W* denotes the linear blend skinning function. Since our task is only to predict human pose and IMU data do not contain human morphological features, we removed the shape-blendshape and pose-blendshape, i.e., all limb lengths in the dataset are equal.

### 3.2. Training Data

The FDIP requires the user to wear inertial sensors in six areas: hips, left lower leg, right lower leg, left lower arm, right lower arm, and head to capture the whole body posture. We need an IMU sensor data (acceleration and rotation) to train our PoseNet. Although the DIP-IMU dataset provides these data, the amount of data is minimal and insufficient to train a model with acceptable generalization. Hence, we need a dataset that includes other action categories. We synthesize inertia data for only six joints in this section.

We utilized the AMASS [19] motion dataset, which is a collection of existing motion capture (mocap) datasets comprising pose data parameters for over 40 h of different types of motions gathered and executed on more than 300 experimental individuals, including 18 datasets such as ACCAD [20], BMLhandball [21], CMU [22], DanceDB [23], HumanEva [24], and Human3.6 [25]. Each sub-dataset in the AMASS dataset contains rotation and whole-body displacement parameters for each joint of the SMPL model. Although it does not contain IMU sensor data, we were able to obtain virtual IMU data by synthesizing the sensor data, so here we used the same approach as TransPose [3] by placing the virtual IMU on the vertices of the SMPL grid and calculating the acceleration of the IMU at that position according to Equation (Equation 2) and its corresponding rotation.
(2)$$a_{i}\left(t\right)=\frac{x_{i}\left(t-n\right)+x_{i}\left(t+n\right)-2x_{i}\left(t\right)}{{\left(n\Delta t\right)}^{2}},i=1,2\ldots 6.$$
where $a_{i}\left(t\right)$ denotes the acceleration measurement of the *i*-th sensor in the *t*-th frame and Δt specifies the interval between two consecutive frames. Here, we sample all the data at 90 fps, and thus Δt=0.0166. We utilized their approach to synthesize data and discover that n=3 is closest to the actual sensor data. Furthermore, we found that synthetic IMU rotations can directly use the converted global rotation of the pose data provided in the AMASS dataset. The code to synthesize the dataset is included in our open source code for easy reproduction.

### 3.3. Sensor Calibration

Since the inertial sensor collects data from the sensor’s local coordinate system, and the FDIP input is the global rotation of the SMPL [18] model joints, we need to convert the sensor’s local coordinate system into the coordinate system corresponding to the SMPL [18] character’s skeleton.

We define the sensor local coordinate system as $F_{w}$ and the SMPL [18] coordinate system as $F_{m}$, corresponding to the base matrix of $B_{w}$ and $B_{m}$, respectively. First, choose any sensor to coincide with its coordinate system with the SMPL [18] character coordinate system (as shown in Figure 1). At this moment, the sensor’s direction (*Q*) is changed from $F_{m}$ to $F_{w}$
(3)$$B_{w}=B_{m}Q.$$

Next, we wear all the sensors onto their corresponding bones. To achieve a stable calibration pose, we have the experimenter stand in a specified pose (T-pose) for a few seconds (Figure 1) and take the average of the pose as the calibration data. Since the sensors can be placed arbitrarily on the body joints, there is a bias of $R_{w}^{offset}$ between the sensors and the joints, and we assume that the sensor location will not be moved during the experiment. We can acquire
(4)$$R_{w}^{bone}=R_{w}^{sensor}R_{w}^{offset}.$$
where $R_{w}^{sensor}$ is the sensor’s reading in the coordinate system $F_{w}$.

Since we let the experimenter execute the calibrated pose, the absolute orientation of the skeleton is the same in $F_{w}$ and $F_{m}$, giving:(5)$$B_{m}R_{m}^{bone}=B_{w}R_{w}^{bone}.$$

The direction of the character bones in the coordinate system $F_{m}$ is represented by $R_{m}^{bone}$, while the direction of the character bones in the world coordinate system is represented by $R_{w}^{bone}$.

The formulas of (3), (4) and (5) may be combined to determine the orientation of the bones in the model:(6)$$R_{m}^{bone}=Q^{-1}R_{w}^{sensor}R_{w}^{offset}$$
(7)$$a_{m}^{bone}=Q^{-1}\left(a_{w}^{sensor}-a_{w}^{offset}\right)$$
where the $a_{w}^{offset}$ is used to eliminate the effect of gravitational acceleration. $R_{m}^{bone}$ and $a_{m}^{bone}$ are used as inputs to the model. We use the rotations and accelerations of the six joints after calibration.

## 4. Method

In this section, we describe our model structure, training and post-processing methods. A schematic of the whole process is shown in Figure 2.

### 4.1. Normalization

We use Equations (6) and (7) to calculate the global joint rotation and local acceleration data, using the joint rotation and acceleration as inputs to our neural network. It is challenging to learn the rotation relationship between joints, we use the same normalization method as DIP [2] and TransPose [3], and we normalize the data input to the network relative to the spine joints (root joint). Here, we name the joints other than the span as the leaf joints. The method is as follows:Normalize the leaf joint rotation relative to the root
(8)$$\begin{array}{c} a_{\mathrm{leaf}\;}=R_{\mathrm{root}\;}{}^{-1}\left(a_{\mathrm{leaf}\;}-a_{\mathrm{root}\;}\right)/30\\ R_{\mathrm{leaf}\;}=R_{\mathrm{root}\;}^{-1}R_{\mathrm{leaf}\;} \end{array}$$Normalize the acceleration of the root
(9)$$\begin{array}{c} a_{\mathrm{root}\;}=R_{\mathrm{root}\;}^{-1}a_{\mathrm{root}\;}/30 \end{array}$$
where $R_{root}$ denotes the rotation of the root node, $R_{leaf}$ denotes the rotation of the leaf joint, $R_{leaf}$ denotes the rotation of the leaf joint, and $a_{leaf}$ denotes the acceleration of the leaf joint. The 30 here is to adapt the acceleration to the network input and adjust the acceleration data size. This scaling factor of 30 is from the TransPose implementation, which is an empirical value.

### 4.2. Pre-Training and Fine-Turning

In this section, we introduce our method for predicting pose. It is relatively challenging to discover the mapping of the 15 joints of the full body from sparse inertial data. Sparse inertial data typically create uncertainty in motions, such as standing and sitting, when the IMU’s acceleration and rotation data are virtually the same. We solve the above problem using a priori knowledge of human poses and temporal information. Figure 2 shows our pose estimation framework. DIP [2] uses the bidirectional LSTM [17] to predict joint rotation directly, and TransPose [3] uses a multi-stage bidirectional LSTM to predict rotation. We present our method to continue the idea of previous work.

First, we perform the pre-training phase task to train PoseNet using the AMASS dataset. Then, we pass the empirical knowledge of the pose to the network in the fine-turning phase to fine-turn PoseNet using the DIP-IMU [2] dataset. Furthermore, we use the network in the fine-turning stage as the final online pose prediction model.

### 4.3. Faster Deep Inertial Pose(FDIP)

The input to FDIP is $x=[a_{lArm},a_{rArm},\cdots ,a_{root},R_{lLeg},R_{rLeg},\cdots R_{root}]$, where *a* and *R* are obtained by Equation (Equation 8) and (9), *R* is the 6d rotation [26], and the 6d rotation has fewer parameters than the 9d rotation matrix. We utilize biRNN [17] and SRU [27] to learn predictions from six IMUs measurements to 15 joint rotations of the whole body. Due to the highly sparse data, the IMU data acquired are essentially identical despite the experimenters conducting entirely distinct actions. In this scenario, it is pretty helpful to examine the use of temporal information to clarify this uncertainty. Furthermore, future data will benefit prediction, so we utilize biRNN. DIP and TransPose both use biLSTM [28] for prediction, but due to the serial structure of LSTM, it restricts the training and inference speed of RNN, which is not parallelizable compared to CNN. In contrast, the SRU network structure can process large operations in parallel and small operations serially.

We slice each action into data of a length of 300 frames. With less than 300 frames, we round off. Considering the DIP dataset contains acceleration data, there may be jitter and noise. Therefore, we present the set of AMASS [19] data used for pre-training into acceleration, adding Gaussian noise.

The first linear layer of PoseNet (Figure 3) is used to extract the features of the current action from the data of the six IMUs, and its hidden units are set to 256. In the second module, we stack five bi-directional SRU layers to extract time-dependent information from past and future action data and predict 15 joint action feature vectors, and the hidden units of each SRU layer are set to 256. SRU is a recurrent neural network similar to LSTM, capable of parallel computation such as a convolutional neural network, which greatly improves the training and evaluation speed. We add layer normalization and dropout to the SRU layer to enhance the generalization performance. The third linear layer is utilized to recover the 15 joint action feature vectors from the 6d rotation data, and its hidden units are set to 512. Figure 3 shows the structure of our pose prediction network.

Finally, SmoothL1Loss [29] is used as the loss function of the training network. SmoothL1Loss makes the loss more resistant to aberrant data in the IMU data, which is insensitive to abnormal data and may regulate the amount of gradient compared to the L2 loss function, which is described as
(10)$$\begin{array}{cc} & \mathrm{Loss}\left(\tilde{\theta },{\tilde{\theta }}^{GT}\right)=\\ & \frac{1}{n}\sum_{i=1}^{n}\begin{array}{cc} \; & 0.5^{*}{\left({\tilde{\theta }}^{GT}\left(t\right)-\tilde{\theta }\left(t\right)\right)}^{2},if\left|{\tilde{\theta }}^{GT}\left(t\right)-\tilde{\theta }\left(t\right)\right|<1\\ \; & \left|{\tilde{\theta }}^{GT}\left(t\right)-\tilde{\theta }\left(t\right)\right|-0.5,\;\mathrm{others}\;. \end{array} \end{array}$$
$\tilde{\theta }$ is our predicted 15 joint rotations and ${\tilde{\theta }}^{GT}$ is the 15 joint rotations from the dataset. We use Equation (Equation 10) to optimize the model so that the predicted results are close to the ground true joint rotations.

### 4.4. Data Compensation

Using the PoseNet pose estimation network, we obtained the root’s joint rotation and the predicted global rotation of 15 joints. The remaining eight joints’ parents can drive the rotation of the remaining eight joints, so here we used the unit rotation matrix instead of the rotations of these eight joints. Combining the above three parts of the rotation, and finally, the local rotation of all joints of the whole body according to the hierarchical order of the joint tree, i.e., the pose, as
(11)$$\begin{array}{c} \theta =\left[R_{0},stn\left(R_{i}^{\left(6D\right)}\right),R_{j}\right],\\ i=1,2,3,4,5,6,9,12,13,14,15,16,17,18,19;\\ j=7,8,10,11,20,21,22,23. \end{array}$$

Here, the stn function turns the 6d rotation into a rotation matrix, where *i* and *j* represent the index of the predicted joints (Figure 1) and the index of the joints using the unit matrix, respectively.

### 4.5. Jitter Error Reduction

The FDIP model surpasses TransPose in angular inaccuracy but causes considerable jitter. We used real-time filtering to decrease the model jitter without making the model more complicated or impacting the inference performance of the model. We added a phase of real-time low-pass filtering to reduce jitter as follows:(12)$$\hat{\theta }\left(t\right)=0.6\tilde{\theta }\left(t-1\right)+0.4\tilde{\theta }\left(t\right).$$

The 0.4 and 0.6 parameters in Equation (Equation 12) come from our experimental results, and their application gives the least jitter.

### 4.6. Training Detail

All training and testing were completed on a computer equipped with an Intel(R) Core(TM) i7-9700K CPU and an RTX 2080Ti graphics card. The model was trained using CUDA 11.1 and Pytorch 1.10.1. The live demo client is implemented using the Unity engine, and the sensors used Noitom Legacy IMU sensors to collect data. We use the Adam [30] optimizer to optimize the loss function Equation (Equation 10). The batch size is set to 128, the learning rate is set to 0.001, and each experiment is trained for 100 epochs and tested with the same random seed, which is set to 42. The PoseNet module was trained using the AMASS [19] dataset into inertial data to obtain a priori knowledge of the movements in the dataset. Then, the DIP dataset was used in the pre-training phase to fine-tune the optimized model better to fit the data from real IMU sensors. The complete code will be provided as Appendix A.

## 5. Experiments

This section evaluates our method and compares it with SIP, DIP, and TransPose, where TransPose is currently the most advanced method for estimating human pose from sparse IMU data. We evaluate these methods using five metrics: SIP Err, SIP Err, Pos Err, Mesh Err, and Jitter Err.

### 5.1. Offline and Online Experiments

Quantitative comparison. Table 1 presents the quantitative comparisons offline, while Table 2 shows the quantitative comparisons online. We offer each metric’s mean and standard deviation offline and online using DIP, SIP, and TransPose. As the table illustrates, our method is similar to TransPose in the offline environment but is three times faster than TransPose in inference and surpasses TransPose in the online situation for all measures. We attribute our advantage to the ideal strategy found by completing many ablation experiments. The input rotation is replaced with a 6d representation rotation, improving the neural network prediction. Our FDIP uses a one-stage model for faster inference, the loss function uses SmoothLoss for insensitivity to outliers, and synthetic acceleration is more similar to the real acceleration value.

Qualitative comparison. Here, we can compare the attitude assessment’s offline and online outcomes more intuitively between our FDIP and the TransPose. Since our method reconstructs the rotation of only 15 joints, to visualize the SMPL pose, we used Equation (Equation 11) to complement the indeed joints to obtain the complete rotation of 24 joints. Figure 4 shows some selected frames from both datasets. From all the examples, it can be observed that our method skillfully reconstructs the hand motion, while TransPose does not correctly estimate the hand motion. Since the IMU data of raised and raised hands are too similar, TransPose does not learn this ambiguity very well.

### 5.2. Dataset Synthesis

To train the model with good generalization, we need more IMU data, so we use Equation (Equation 2) (a smoothing factor of n=4) to synthesize IMU data in the AMASS [19] dataset. However, we find that a smoothing factor of n=3 is closer to the actual sensor data.

To prove the smoothing factor, we re-synthesized the acceleration of the DIP-IMU data set using ground true pose and global displacement. We compared the synthesized acceleration with the real IMU measurement. We evaluated the correct acceleration percentage at some selected error thresholds and plotted the curve and error distribution in Figure 5. The results show that the smoothing factor of n=3 produces acceleration similar to that of real sensors. Therefore, we used n=3 in our work.

Figure 6 shows the data waveforms of the actual orientation measurements extracted from the DIP real dataset and the synthesized orientation measurements. The direction measurements can be directly used from the global rotation data of the pose without synthesis.

### 5.3. 6d and 9d Rotation

The DIP and TransPose proposed methods using IMU measurements (acceleration and 9d rotation) as input and 9d rotations of joints as output. We altered the input to 6d rotations. Since 6d rotation has fewer parameters and better continuity in deep learning than 9d rotation, we individually examined the training with 6d rotation and 9d rotation. We determined that there is not much difference between them. However, since 6d rotation has a lower number of parameters, subsequent studies were performed using 6d rotation representation. Table 3 shows the results of our rotation with 6d and 9d.

### 5.4. Effect of Acceleration

Using rotation or acceleration alone, as described in SIP, produces significant errors, and we experimented with removing the effect of acceleration in FDIP. Considering that acceleration in our experiments does reduce the error rate but increases jitter, this jitter is negligible. Table 4 indicates the effect of our acceleration characteristics on the outcomes.

### 5.5. Effect of Filter

We used Equation (Equation 12) to fuse the rotations of the previous and current moments, and this method not only achieves jitter elimination but also does not lead to a more complex model. The experimental results are shown in Table 5, where the real-time low-pass filtering effectively decreases the jitter of the model.

### 5.6. Inference Speed

We utilized a typical bidirectional recurrent neural network with SRU. The rationale for using SRU instead of LSTM is that LSTM units have to operate the samples sequentially to output the results. This activity prohibits the unit from being most helpful in an environment where numerous computers compute in parallel. The SRU unit was published in 2017. This keeps the cyclic structure of the LSTM unit. It enhances the speed of operations by altering the sequence of operations (placing matrix multiplication outside the serial loop and putting the multiply-and-add operations within the serial circle). Our model does not use neural networks to predict the support leg and uses a one-stage SRUCell pose prediction. On CPU and GPU, our online pose estimation is 2.5 times faster than the TransPose model with the displacement estimation removed. Online inference speeds up to 90 FPS. Our offline pose estimation is eight times faster than the TransPose model.

## 6. Limitations

We need to use inertial sensors to acquire attitude data on the hardware side. However, the influence of the ambient magnetic field can lead to inaccurate and drifting acquired attitudes. Therefore, we need to conduct online experiments in an environment with low magnetic field interference. The accuracy of the inertial sensors also dramatically influences the prediction. We use the Noitom Legacy to acquire data more accurately than the inexpensive sensor MPU9250.

On the software side, the IMU can accumulate drift over a long execution period, leading to increasingly poor pose estimation. Since IMU does not include position measurement, visual methods can easily measure position. The sparse inertial poser can add the VR headset, and its pose and displacement can be observed by sensors such as cameras or LIDAR carried on the headset. We believe that this will be the future direction of development. On the other hand, the latency of the whole real-time pose estimation system lies above the network communication; if we use C# to implement the whole system, we believe that we can further reduce the latency, and we will continue our research in this area in the future.

## 7. Conclusions

We present FDIP, which captures the whole body pose using only six sensors, with 250% faster online inference and 16% less angular error than the state-of-the-art work. In this technique, we perform ablation experiments on previous methods and fuse their best to propose our model, including an optimized acceleration synthesis method using 6d rotation, Bi-directional SRU, and SmoothLoss. FDIP can facilitate wearable devices in the home and will play a vital role in the metaverse.

## Figures and Tables

**Figure 1 sensors-22-07144-f001:**
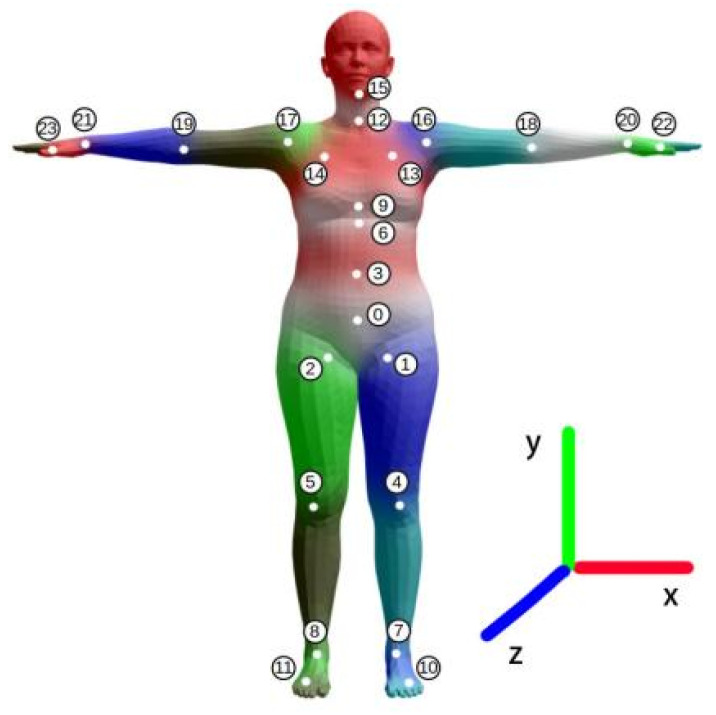
SMPL parametric model.

**Figure 2 sensors-22-07144-f002:**
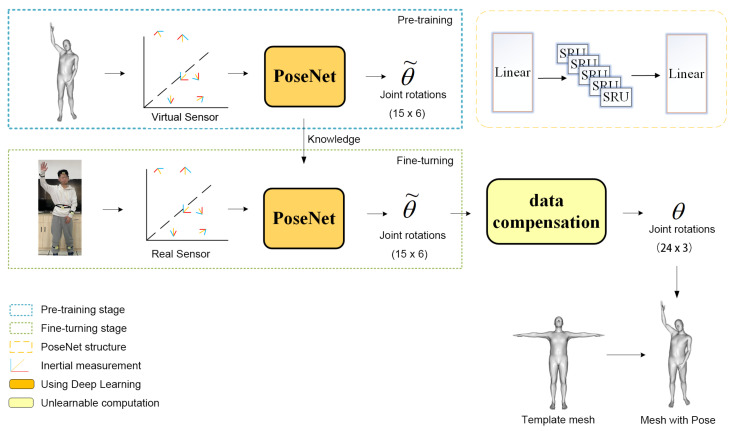
Overview of our pipeline. We make pose reconstruction a one-stage task and feed the IMU measurement data directly into the neural network, retrieve the 6d rotations of the 15 joints, and then augment the rotations of the remaining joints. Finally, we use real-time filtering to decrease jitter in motion capture.

**Figure 3 sensors-22-07144-f003:**
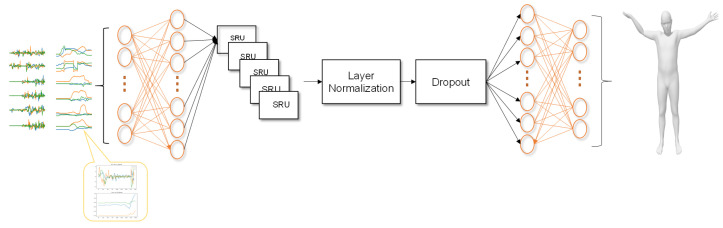
The structure of PoseNet. Inputs of acceleration and rotation after normalization from IMU to predict the rotation of 15 joints.

**Figure 4 sensors-22-07144-f004:**
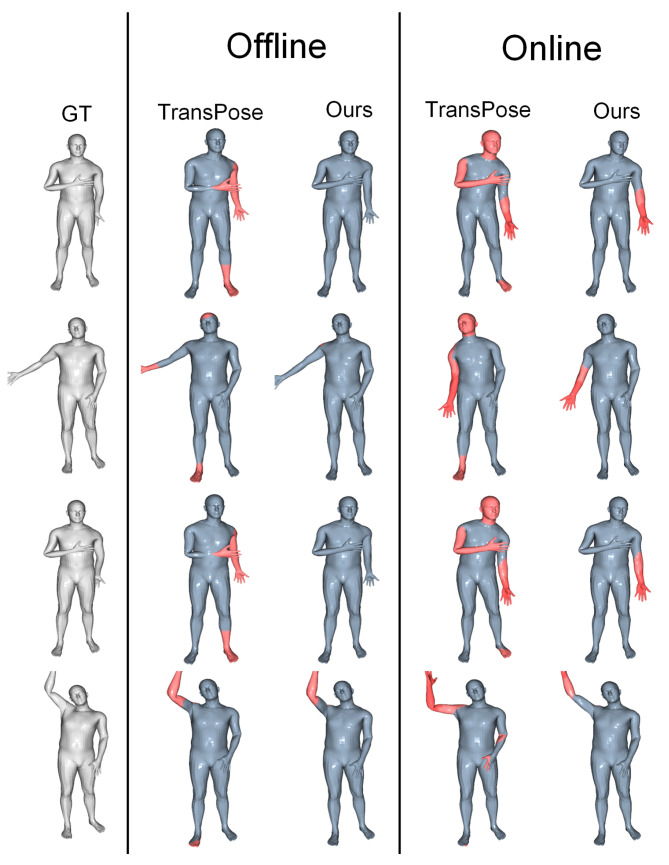
A qualitative comparison between our approach, and TransPose. We performed offline and online comparisons on the DIP-IMU dataset, and the chosen findings are shown below. We included the comparison video among the Appendix A. Each vertex is colored based on its location from where it should be on the ground. The greater the error is, the more prominent the red marker.

**Figure 5 sensors-22-07144-f005:**
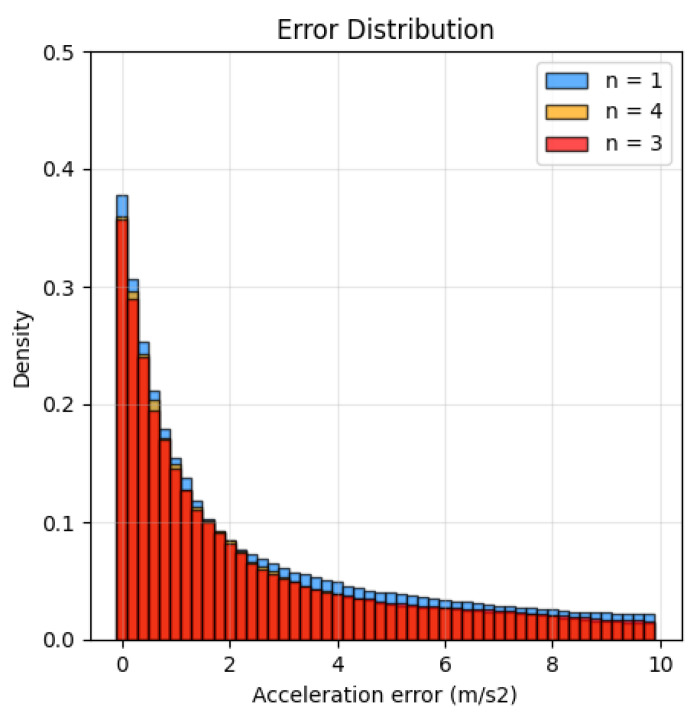
The acceleration error also represents the Manhattan distance between the acceleration of the same actual data frame and synthetic data. Density describes the percentage of frames in the whole sequence where the difference between real and synthetic data is more significant than the error threshold.

**Figure 6 sensors-22-07144-f006:**
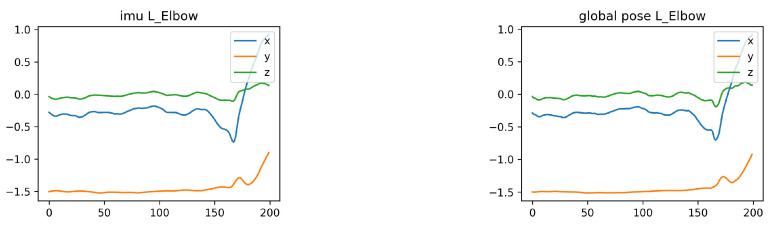
DIP for real and synthetic IMU measurements.

**Table 1 sensors-22-07144-t001:** Comparison of offline experimental results. SIP Err: global average rotational measurement error of the angles of the upper arms and thighs. Ang Err: global average rotational error of the full-body joint points. Positional Err: the average Euclidean distance error measured in centimeters for all estimated joints with root joint (spine) alignment. Mesh Err: measures the average Euclidean distance error in centimeters for all vertices of the estimated body mesh aligned with the root joint (Spine). The vertices can be calculated according to Equation (Equation 1). Jitter Err: measures the average jitter of all body joints in the predicted motion. Jerk is the third-order derivative of position for time and reflects the smoothness and naturalness of the motion. A more negligible average jitter implies a smoother and more realistic animation.

	SIP Err (deg)	Ang Err (deg)	Pos Err (cm)	Mesh Err (cm)	Jitter Err ($10^{2}$m/s${}^{3}$)
SIP	21.02 (±9.61)	8.77 (±4.38)	6.66 (±3.33)	7.71 (±3.80)	3.86 (±6.32)
DIP	16.36 (±8.60)	14.41 (±7.90)	6.98 (±3.89)	8.56 (±4.65)	23.37 (±23.84)
TransPose	13.97 (±**6.77**)	7.62 (±**4.01**)	4.90 (±**2.75**)	**5.83 (±3.21**)	**1.19 (±1.76**)
Ours	**13.38** (±6.87)	**7.50** (±4.10)	**4.86** (±2.82)	5.84 (±3.31)	3.59 (±4.80)

**Table 2 sensors-22-07144-t002:** Comparison of the online experimental results.

	SIP Err (deg)	Ang Err (deg)	Pos Err (cm)	Mesh Err (cm)	Jitter Err ($10^{2}$m/s${}^{3}$)
DIP	17.10 (±9.59)	15.16 (±8.53)	7.33 (±4.23)	8.96 (±5.01)	30.13 (±28.76)
TransPose	16.68 (±8.68)	8.85 (±4.82)	5.95 (±3.65)	7.09 (±4.24)	6.11 (±7.92)
Ours	**13.92 (±7.90)**	**7.60 (±4.38)**	**5.18 (±3.27)**	**6.17 (±3.76)**	**5.69 (±6.72)**

**Table 3 sensors-22-07144-t003:** Evaluation results for 9d rotation and 6d rotation.

Method	SIP (deg)	Ang (deg)	Pos (cm)	Mesh (cm)	Jitter ($10^{2}$m/s${}^{3}$)
9d and acc	13.14 (±6.79)	7.02 (±3.73)	4.71 (±2.68)	5.61 (±3.13)	1.01 (±0.84)
6d and acc	13.11 (±6.71)	6.96 (±3.71)	4.87 (±2.78)	5.80 (±3.24)	1.10 (±0.89)

**Table 4 sensors-22-07144-t004:** Impact of using acceleration as a feature on the evaluation results.

Method	SIP (deg)	Ang (deg)	Pos (cm)	Mesh (cm)	Jitter ($10^{2}$m/s${}^{3}$)
FDIP	15.25 (±8.95)	7.69 (±4.70)	5.38 (±3.47)	6.25 (±3.94)	2.85 (±11.48)
FDIP and acc	13.38 (±6.87)	7.50 (±4.10)	4.86 (±2.82)	5.84 (±3.31)	3.59 (±4.80)

**Table 5 sensors-22-07144-t005:** Impact of using the filter on the evaluation results.

Method	SIP (deg)	Ang (deg)	Pos (cm)	Mesh (cm)	Jitter ($10^{2}$m/s${}^{3}$)
FDIP	13.20 (±6.79)	6.90 (±3.70)	4.71 (±2.75)	5.61 (±3.21)	10.92 (±16.09)
FDIP & filter	13.38 (±6.87)	7.50 (±4.10)	4.86 (±2.82)	5.84 (±3.31)	3.59 (±4.80)

## Data Availability

Restrictions apply to the availability of these data. Data was obtained from the Max Planck Institute for Intelligent Systems and are available [https://amass.is.tue.mpg.de/] with the permission of the Max Planck Institute for Intelligent Systems.

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
