# Peer review of "Faster Deep Inertial Pose Estimation with Six Inertial Sensors"

_sensors, 2022, doi:10.3390/s22197144_

Round 1

Reviewer 1 Report

The paper proposes a Deep learning network for human posture estimation 3D/3D mesh. I have the following comments and questions.

-      Figure 3 does not show what is the input of the human posture estimation network? Color image or depth image?

- In Figure 1, 24 points are used to represent the parameters of the human model. However, the transformations in other parts (3.2. Training Data, ) use 15 points?

-      To train our PoseNet, we need IMU sensor data (acceleration and rotation).
Although the DIP-IMU dataset provides these data. Is the PoseNet network recommended by you?

-      - Figure 2 font is small and inconsistent.

-      “The inertia data we input to the neural network are global joint rotations(Please see 172
Appendix A??”,  not found  Appendix A

-      In Figure 2, what is SRU? You should explain it

-      It is challenging to learn the rotation relationship between joints, we refer
to DIP[
2] and TransPose[1], and we normalize the data input to the network relative to the
spine joints (root joint). The combination of DIP[2] and TransPose[1] is not shown?

- In formula (8), (9) why divide by 30, “The 30 here is to adapt the acceleration to the network input and adjust the acceleration data size”. Why would you choose 30 and not 20 or 40, etc. A detailed analysis of this issue is required.

- Figure 3 has a lot of unclear information, and at the same time it is distorted.

What is GT in formula (10)? “The superscript gt is the ground-true of the 15 joint rotations.” This sentence has no meaning.

- In formula (12) why set scale as 0.6 and 0.4

- There are many parameters in the item “4.6. Training Detail” not presented as batch-size, number epoch, learning rate, etc.

- The evaluation measures in Tables 1 and 2 should be built uniformly.

What is 102m/s3 in tables 1 and 2?

- The symbols for +/- in table 1,2,3,4,5 should be unified.

-      - 90fps information is not found in the results tables

-       

-      - Figure 4,5,6 are too big compared to other pictures

-       

- What is N=1,3,4 in figure 3

- It is very important that the method you propose has not been compared with any human pose or mesh human pose conventions that have been studied as below.

https://akanazawa.github.io/hmr/

https://www.sciencedirect.com/science/article/abs/pii/S1051200422002457

https://link.springer.com/article/10.1007/s11227-021-04184-7

- The source code of the proposed method should be made public.

- Information and description of AMASS[18] dataset has not been presented

- There are still a lot of typos in the article.

- The related works section should cite the article:

https://www.mdpi.com/1424-8220/22/14/5419

Reviewer 2 Report

THis paper presents an interesting study which will bw of interest to the intended audience. The M&M are described with the results.However, I have comments which must be addressed in a revised version:

1) While the manuscript has a descriptive title and abstract the use of acronyms must be restricted to the main body of the text with all acronyms defined on first use. The title would be improved is acronyms are removed.

2) The keywords are inadequate and must be revised and extended to introduce apprpriate index terms.

3) The introduction requires a paragraph setting out the paper structure.

4) The authors must reference all the equation numbers in the text.

5) In Section4.1 (line 173) reference is made to "Appendix A ??". What is this reference and the related information. This must be addressed.

6) The formatting of the manuscript must be improved. For example, see Figure 3 and Tables 3 and 4 (column overflow).

7) The treatment of limitations needs improvenent with a clearer discussion around the open research questions and related directions for future research.

8) I found the the paper failed to properly consider the practical managerial significance and application in the 'real-world'. This is required.

In summary, this ia potentially an interesting paper thar needs revision to address the comments set out in my review.

Reviewer 3 Report

It is recommended to revise the Figure 5. including the appropriate descriptions of the procedure.

Round 2

Reviewer 1 Report

My questions and comments were answered in the revised version of the article

Reviewer 2 Report

I have read the authors response and the revised manuscript. In general I am content that the response and revisions have addressed the concerns expressed in my review.